# A Privacy-Preserved Internet-of-Medical-Things Scheme for Eradication and Control of Dengue Using UAV

**DOI:** 10.3390/mi13101702

**Published:** 2022-10-10

**Authors:** Amir Ali, Shibli Nisar, Muhammad Asghar Khan, Syed Agha Hassnain Mohsan, Fazal Noor, Hala Mostafa, Mohamed Marey

**Affiliations:** 1Military College of Signals (MCS), National University of Sciences and Technology, Islamabad 44000, Pakistan; 2Department of Electrical Engineering, Hamdard University, Islamabad 44000, Pakistan; 3Smart Systems Engineering Laboratory, College of Engineering, Prince Sultan University, Riyadh 11586, Saudi Arabia; 4Ocean College, Zhejiang University, Zheda Road 1, Zhoushan 316021, China; 5Faculty of Computer and Information Systems, Islamic University of Madinah, Madinah 400411, Saudi Arabia; 6Department of Information Technology, College of Computer and Information Sciences, Princess Nourah bint Abdulrahman University, P.O. Box 84428, Riyadh 11671, Saudi Arabia

**Keywords:** UAV, dengue, internet of medical things, privacy, tracking systems

## Abstract

Dengue is a mosquito-borne viral infection, found in tropical and sub-tropical climates worldwide, mostly in urban and semi-urban areas. Countries like Pakistan receive heavy rains annually resulting in floods in urban cities due to poor drainage systems. Currently, different cities of Pakistan are at high risk of dengue outbreaks, as multiple dengue cases have been reported due to poor flood control and drainage systems. After heavy rain in urban areas, mosquitoes are provided with a favorable environment for their breeding and transmission through stagnant water due to poor maintenance of the drainage system. The history of the dengue virus in Pakistan shows that there is a closed relationship between dengue outbreaks and a rainfall. There is no specific treatment for dengue; however, the outbreak can be controlled through internet of medical things (IoMT). In this paper, we propose a novel privacy-preserved IoMT model to control dengue virus outbreaks by tracking dengue virus-infected patients based on bedding location extracted using call data record analysis (CDRA). Once the bedding location of the patient is identified, then the actual infected spot can be easily located by using geographic information system mapping. Once the targeted spots are identified, then it is very easy to eliminate the dengue by spraying the affected areas with the help of unmanned aerial vehicles (UAVs). The proposed model identifies the targeted spots up to 100%, based on the bedding location of the patient using CDRA.

## 1. Introduction

The history of the dengue virus in the human population is not known because the dengue virus is asymptomatic, and the diagnosis of such diseases is difficult [1]. In 992 BC, the Chinese medical encyclopedia recorded the earliest data of the dengue virus [2]. At the end of the 18th century, there were other epidemic diseases in America and Asia. These epidemic diseases were very similar to the dengue virus. Therefore, based on these observations, the hypothesis is presented that the dengue virus spread in the human population in the middle of the 19th to 20th century [3]. Humans are infected by the dengue virus through a mosquito bite. One of the families of mosquitoes Aedes aegypti spread this virus, and female mosquitoes transmitted the dengue virus through their bites [4]. The dengue mosquitoes have a very short life, but as their eggs are converted into larvae, on average from 4 to 10 days, they spread the dengue virus [2,5,6]. The most favorable temperature for their breeding is 30 ${}^{\circ }$C. The dengue outbreaks are enhanced if there is rainfall at a high temperature [7]. Dengue fever presents symptoms such as body pain, nausea, vomiting, severe headache, and many more [4,8]. At the early stage of dengue fever, it has the same symptoms as common fever, but these symptoms become intense as time passes. There are four types of dengue virus, DENV1, DENV2, DENV3, and DENV4 identified to date [5]. In countries like Pakistan, the rate of dengue virus spread is very difficult to control because of poor drainage and solid wastes systems. In Pakistan, the monsoon period starts from July to October; during this period, there are heavy rains, and most populated areas are badly affected by floods. As there is an obsolete system of flood control and drainage system, there is a high risk of water diseases, such as dengue and malaria [9]. The spread of the dengue virus is directly linked with monsoon rainfalls in Pakistan, as this country receives heavy rainfalls annually in this season, and due to poor drainage systems and infrastructure, most cities are heavily affected by the flood [9]. Aedes mosquitoes easily breed in stained water, and this ideal environment is provided on rooftops and water containers where stained water gathers after each episode of rainfall [9]. In 2010, there was a heavy rainfall resulting in floods in Pakistan, which directly affected 14–20 million people in different cities [10]. As a result of these floods, Aedes mosquitoes had a highly favorable environment for breeding, resulting in an enormous increase in cases of dengue virus [11]. Until then, these were the highest recorded cases of the dengue virus in Pakistan. These rainfalls are one of the most important factors for dengue virus spread as shown in Figure 1 [12].

Other factors are densely populated cities, unhygienic drinking water, and refugee camps, which enhance dengue virus spread in different regions of the country. Dengue virus was first set up in Pakistan by importing tires with eggs of the infected mosquitoes in them at the seaport of Karachi [13]. In 1994, the first outbreak of the dengue virus occurred, targeting the southern region of Pakistan in Karachi city [14]. From the study, it was confirmed that both types of dengue virus, DENV-1 and DENV-2, were introduced in the first outbreak [15]. The third type of dengue virus DENV-3 was confirmed in the outbreak that occurred in 2005–2006 in Karachi [16,17]. From 2007 to 2009, Lahore was majorly affected by serotypes, DENV-2 and DENV-3 [18]. In 2010, a massive outbreak occurred, affecting Khyber Pakhtunkhwa, Punjab, and Sindh; the main factor of this large-scale outbreak was floods which occurred in the month of July that year [11].

This paper proposes a novel geographical position-based model to control dengue virus outbreaks by tracking and extracting the bedding location of dengue virus-infected patients through CDRA while keeping the privacy of the patient intact. After the extraction of bedding locations, these bedding locations are mapped or spotted on GIS mapping using IoMT-based cloud computing. Then an algorithm initiates to find the number or percentage of reported cases in these bedding location spots. This algorithm sets a threshold if the number of reported cases exceeds the threshold, then that location is marked red on the GIS map. Information about these red-marked locations is shared from the cloud to the concerning health department, and then the department is responsible for sending spray drones to red-marked locations.

## 2. Related Work

Health records in developing countries need to be digitalized. Hospitals record data of reportable diseases using electronic health records (EHR). Hospitals use different EHR systems; they need to have standard EHR systems installed to record and maintain cases of reported diseases. Public health informatics contribute their work in the local area, reporting cases using techniques that are not of standard form [19,20,21]. To control dengue, it is important to establish a real-time case-reporting mechanism in a specific region rather than passive reporting. As dengue is a reportable disease and in the context of its surveillance the data collection from the regional to national levels remain fragmented and not in a standard format, the same mechanism is followed in many other developing countries [22,23]. Health Level Seven (HL7) is a standard that is not only recognized internationally for data exchange in clinical departments, but also used by the management of public health in the surveillance of diseases [24,25,26,27,28,29,30,31,32,33]. However, in the literature, there are few examples where the HL7 standard is used for improving the passive surveillance of reportable diseases, such as dengue [34,35,36].

Several factors of dengue outbreaks and their epidemiology have been explored in literature [37,38]. There are many factors which enhance dengue spread; climatic and non-climatic factors are investigated for the prevention and control of dengue disease in populations that are more prone to dengue spread. The following steps are followed in the process: dengue hotspots are identified as well as the relationship between dengue fever incidences and its transmission that accelerates dengue fever spreads globally [39]. Other aspects of the dengue virus have been studied with several factors, including geographical, climatic, topographical, migration, demographic, and traveling [40,41,42,43,44]. An effective model is built for the prediction of dengue virus spread, and an AI-based analysis of factors which are strongly correlated with the rate of disease incidence is conducted.Factors that have a strong impact on dengue transmission have been reported in several studies, such as rainfall and temperature [45,46,47]. Many reasons have been given to support the rainfall and temperature factors. For example, mosquitoes survive in warmer temperatures, which provides favorable conditions for their growth, reproduction, and transmission rates. Moreover, rainfall provides favorable sites for their breeding, which has a direct impact on the rate of disease incidence [48,49]. However, in some studies, this correlation has been negated by providing the reason that, due to heavy rainfall, mosquito breeding sites are being removed [50,51]. Some studies have shown that after hot and dry weather, when there is heavy rainfall, dengue fever cases increase [52,53,54].

Therefore, dengue virus transmission and its relationship with climatic factors are not simple and need further research on the dynamics of dengue in the region of the population with high rates of dengue infections to understand dengue transmission globally. To manage infectious diseases such as, dengue spatial analysis using geographic information system (GIS) has gained its worth in many studies [55,56,57]. In the literature, to study the dengue dynamics in terms of regions and different factors, several mathematical models were presented along with analysis based on AI and many other techniques [58,59,60,61,62]. However, some studies use spatial analysis on a wide scale. Many studies narrow their scale by targeting specific cities, districts, and town levels [63,64,65], while some research works analyze the point time series data to identify the pattern of dengue outbreaks [66,67]. Research contributions on the experimental demonstration of UAVs for dengue control is shown in Table 1.

In Pakistan, dengue cases and their outbreaks put an enormous burden on the government annually [75,76]. The dengue virus outbreaks are caused by many factors, including climatic changes, such as flooding, and natural disasters, such as earthquakes [77,78]. As the dengue virus is endemic in Pakistan, researchers presented their work to prevent and control the disease spread by proposing a spatial–temporal dynamic model [79,80,81]. CDRA and the contact-tracing technique is proposed in this work to control outbreak of the coronavirus. Using this technique, a coronavirus infected individual can be traced. The complete path of the infected individual can be traversed along with all other phone numbers of individuals who have met with the patient by using this technique [82]. In this work, the author proposed a hybrid model based on advanced technologies, which includes cyber, cellular, and wireless low range technologies. In this work, the call data record of cellular technology is used to trace the infected patient, cyber technology is used for calls on the voice over internet protocol, and wireless low range technology is used for the identification of physical contact with others having no call history [83,84].

Applications of drones have continuously increased in recent years and remain successful in gaining researchers attention. The main reason that they remain the center of interest for researchers is their properties, such as delivery service, in which human direct contact is eliminated (non-human contact delivery). Similarly, other properties are surveillance and spray drones. ‘Drone’ is the commercial name of a UAV (unmanned aerial vehicle). A UAV is an aircraft capable of flying without a human pilot and is used to move products of different types from one place to another place. Basic and advanced are the two subcategories of UAV; the first category drones are monitored and controlled remotely by humans, while the second category drones are controlled and monitored by sensors and LiDAR. In the recent COVID-19 pandemic, it was observed that drone usage in the health department was widely increased. During the COVID-19 pandemic, government health departments used these drones for three major tasks: drone delivery service (for the delivery of vaccines and medicines), surveillance (to keep an eye on the people who are violating quarantine rules), and spray drones (used for spraying the infected area for disinfection).

As researchers have concluded that the life of a COVID-19 virus is sufficiently long on different surfaces to remain infectious, it is very much important to disinfect the infected locations as well. For this purpose, spray drones were manufactured. Spray drones are special drones developed for spraying purposes to disinfect infected areas. These drones were manufactured during COVID-19 pandemic [85]. Several countries, such as the United States, France, Chile, Spain, China, the United Arab Emirates, Philippines, and India, have used drones for the three major tasks as discussed to control or minimize direct human contact with infected individuals or infected surfaces [86]. People have faced a lot of problems in terms of accessing markets for daily usage products and different services during COVID-19 lock downs. Drone delivery services are very much appreciated in the public and private sectors. Health departments also use drone delivery services for the delivery of medical appliances and medicines from medical centers to remote areas. The same delivery service was used in Sweden during the COVID-19 pandemic for taking test samples in large quantities from suspected individuals, thus reducing human interaction, which resulted in controlling the spread rate of the COVID-19 pandemic [87]. Germany also used this drone delivery service for delivering test samples to labs from remote areas during the COVID-19 pandemic [88]. Smart lock down was introduced during the COVID-19 pandemic, in which limited areas were opened for normal daily routine work, making sure to maintain social distance. To make sure that people were maintaining social distances, surveillance drones were used with cameras installed in them [89]. Special drones were used during the COVID-19 pandemic for detecting fever; thermal image technology was used for the detection of COVID-19 patients. This technology was used by Alibaba, the Chinese company, and claimed 96 percent COVID-19 patient detection [90]. The detailed comparison of the UAV applications is shown in Table 2.

## 3. Proposed Model

To locate dengue virus infected patients, an internet-of-medical-things (IoMT) based model is required to overcome the spread of dengue virus. The mentioned model can be achieved by using the history of a patient’s call record. This spread can be reduced by tracking and extracting the bedding location of the dengue virus infected patient through call data record analysis (CDRA). The privacy of the infected patient is preserved, as the information will be only available to the relevant and authorized personnel only.

When the suspected person has symptoms of dengue virus, and he or she is tested positive for dengue virus, then he or she will fill the proposed performa, in which he or she must fill five locations addresses where the patient has visited or been for the last 7 days. These provided bedding locations will be extracted through the call data record (CDR). After the extraction of bedding locations, these locations will be spotted on the map through GIS mapping. Then, our proposed model will compare the number of cases in these mapped locations with the threshold set in our model in a radius of 500 m. Now, the identification of actual or infected locations will be achieved by comparing the number of cases with the threshold, and those extracted locations will be marked red on the map where the number of cases exceeds the threshold. The government authority will be informed about these red marked locations to send spray drones to the infected areas. Dengue virus outbreaks can be controlled or completely stopped by implementing our proposed novel model. The working mechanism of the proposed model is shown in the form of a flowchart as shown in Figure 2.

## 4. CDRA Framework

To locate dengue-infected patients, a geographical position-based model is needed to control or minimize the spread of the dengue virus. To achieve the required model, the patient location can be tracked by using his or her call history and the base station from where the patient mobile receives and send signals. Once the patient location is tracked, the dengue virus spread can be controlled or minimized by spraying the infected area with the help of UAVs, isolating, and monitoring the infected dengue virus patient. If this infected patient is ignored at the initial stage, it will result in the spread of the dengue virus drastically. The CDRA algorithm will be used to locate or track the infected patients by monitoring their call history to control the spread of dengue virus. This algorithm can be easily implemented. By using CDRA, after the patient location is tracked, a spray team will be informed to spray the infected location using UAVs. Our proposed model can minimize or control dengue virus outbreaks. A flowchart of our proposed model and the CDRA (IoMT, UAV, GIS and Cloud) framework graphical abstract is shown in Figure 2 and Figure 3, respectively.

CDRA provides information about the caller and called users in the form of metadata. Metadata comprise a set of data that give detailed information about other data, such as how the cellular system is used by users in various ways. These metadata carry the following information: the location of the caller and called parties, the date and time of call, duration of call, IMEI information, cost of call per minute, and active call location (late, long) from the connected base transceiver station (BTS). The overall algorithm of the proposed model is given is given in Algorithm 1.
**Algorithm 1** Proposed model.**Require:**CDR, present address, permanent address, work place address, visiting address, and λ.1:Perform CDRA to extract bedding location2:All suspect locations will be mapped on GIS’3:**if** Number of cases **then** > λ4:    Spray the infected location5:**else**6:    Ignore7:**end if****Ensure:**Sprayed all infected locations

## 5. Case Study

In this section, based on the hypothesis, a case study is investigated to realize the main targets of the proposed model. In this way, the assessment of the desired results from the proposed model can be verified.

When the suspected patient is tested positive with dengue virus, then the first step is to fill the proposed performa as shown in Table 3. He/she will provide the data for the following attributes, such as name, cell number, current address late–long, permanent address/bedding location late–long, temporary address late–long, workplace location late–long, and visited location late–long. The current address is the location at which the patient is filling the performa. The permanent address or location is the residential or bedding location of the patient. The temporary address is the location where the patient used to live for one or two days per week. The workplace address is the location where the patient used to work for five or six days per week. Finally, the visited address is the location where the patient has visited in the last week. So, after filling these fields in the performa, this performa is kept and maintained on IoMT-based cloud, where the data provided in the performa are analyzed by our proposed model. Using IoMT-based cloud computing, our proposed model performs CDR on the provided locations in performa to extract the bedding location of the patient.

After the extraction of bedding locations, these bedding locations are mapped or spotted on GIS mapping using IoMT-based cloud computing. Then an algorithm initiates to find the number or percentage of reported cases in these bedding location spots. This algorithm sets a threshold of 20 reported cases per 500 m and is labeled as λ. If the number of reported cases exceeds the threshold, then that location is marked red on the GIS map. Information about these red marked locations is shared from the cloud to the concerning health department, and now this department is responsible for sending spray drones to red-marked locations.

### 5.1. Mr. A

Let us assume that an infected patient, named Mr. A, fills the performa by providing the attributes such that his current address, permanent address, temporary address, workplace address, and visited address all are the same location as shown in Table 4. In this case, as all addresses show the same location, the government authority receives information about the infected location from the cloud, and now the department of concern is responsible for sending spray drones to the identified location on the GIS mapping as shown in Figure 4.

### 5.2. Mr. B

In this case, let us suppose that the patient named Mr. B fills the performa by providing the attributes in the following manner such that his current address, permanent address, and workplace address are the same, while his temporary address and visiting address are the same as shown in Table 5. In this case, IoMT-based cloud computing extracts two locations mapped on GIS, and now our proposed algorithm is initiated. One of the two locations has a high density in reported cases; this high-density location is identified by the threshold set in our algorithm, and this location is marked red on the GIS mapping. After that, the government authority receives information about the infected location from the cloud, and now the department of concern is responsible for sending spray drones to the identified location on the GIS mapping as shown in Figure 5.

### 5.3. Mr. C

In our third case, let us suppose that patient Mr. C fills the performa, and his attributes show that his current and permanent addresses are the same and his temporary and visited addresses are the same, while his workplace address is different as shown in Table 6. In this case, three different locations are extracted and mapped on GIS by IoMT-based cloud computing, and the proposed algorithm is initiated based on these three different locations. The number of reported cases in one of the three identified locations exceeds the threshold. This location is marked red on the GIS mapping. The government authority receives information about the infected location from the cloud, and now the department of concern is responsible for sending spray drones to the identified location on the GIS mapping as shown in Figure 6.

### 5.4. Mr. D

The fourth case shows that the patient, Mr. D, fills the performa, and his attributes show that his permanent and workplace addresses are the same, and his current and temporary addresses are the same, while his visiting address is different as shown in Table 7. In this case, again, three different locations are extracted from IoMT-based cloud computing, and the same procedure is followed as in case 3. However, if the number of reported cases in two of the three locations exceeds the threshold, then the GIS mapping marks these two locations in red. The government authority receives information about these two infected locations from the cloud, and now the department of concern is responsible for sending spray drones to these identified locations on the GIS mapping as shown in Figure 7.

### 5.5. Mr. E

In the 5th case, let us assume that for patient Mr. E, his performa attributes show that his permanent and current addresses are the same, while his workplace, visited, and temporary addresses are different as shown in Table 8. As in this case, four different locations are extracted from IoMT-based cloud computing, all these locations are mapped on GIS. The proposed algorithm extracts those locations where the number of reported cases exceeds the threshold and marks these locations red on the GIS mapping. After that, the government authority receives information about these infected locations from the cloud, and now the department of concern is responsible for sending spray drones to these identified locations on the GIS mapping as shown in Figure 8.

In the literature, for the tracking and tracing of infected individuals, different models were proposed, but they were unable to track viral infection spread due to dengue. A comparison of our proposed model with existing tracking models is summarized in Table 9. This comparison in Table 9 shows that this proposed model can track and trace the viral infection spread through a mosquito virus called dengue.

## 6. Conclusions

Dengue has traditionally been categorized as a neglected tropical disease not frequently associated with Pakistan. However, recent dengue activity in Pakistan demonstrates that areas of the country are at risk of experiencing outbreaks. Practitioners in areas with dengue mosquito vector species should be aware of the risk of imported dengue, which could result in an outbreak. Pakistan is one of the underdeveloped countries where there are no proper drainage systems and there is no proper system to keep these drains clear and clean for proper drainage. The government authorities have thus far failed to control dengue virus outbreaks because of improper management in the flood control department. The lack of proper sanitation has provided favorable conditions for the breeding and transmission of mosquitoes, resulting in outbreaks of diseases such as dengue and malaria. Since there is no specific treatment for dengue, this paper has proposed a novel geographical position-based model to control dengue virus outbreaks by tracking and extracting the bedding location of dengue virus-infected patients through CDRA, while keeping the privacy of the patient in tact. After the extraction of bedding locations, these bedding locations are mapped or spotted on GIS mapping using IoMT-based cloud computing. Then, an algorithm is initiated to find the number or percentage of reported cases in these bedding location spots. This algorithm set a threshold of 20 reported cases per 500 m and labeled as λ. If the number of reported cases exceeds the threshold, then that location is marked in red on the GIS map. Information about these red-marked locations is shared from the cloud to the concerning health department, and then the department is responsible for sending spray drones to red-marked locations. The proposed model identified the target spots very accurately based on the bedding location of the patient using CDRA. The information extracted with the help of the proposed model can easily minimize the wide spread of dengue-like epidemics. It is, therefore, recommended that the proposed model be adopted by the government to control the wide spread of dengue. In future, the authors are planning to include different social network data, such as those of Facebook, WhatsApp, Google, Telegram, and Instagram, in the proposed model. This will further improve the identification of target spots in dengue-like epidemics.

## Figures and Tables

**Figure 1 micromachines-13-01702-f001:**
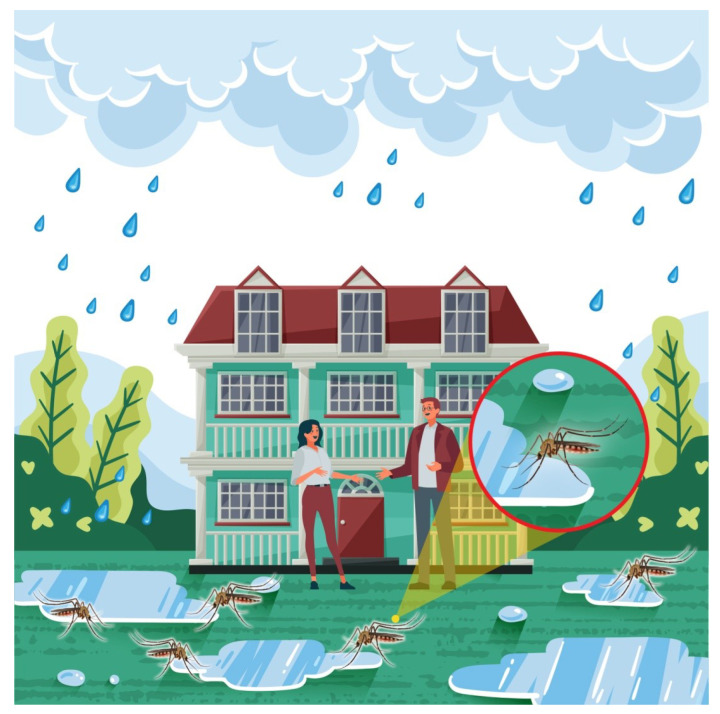
Dengue virus spreading factors.

**Figure 2 micromachines-13-01702-f002:**
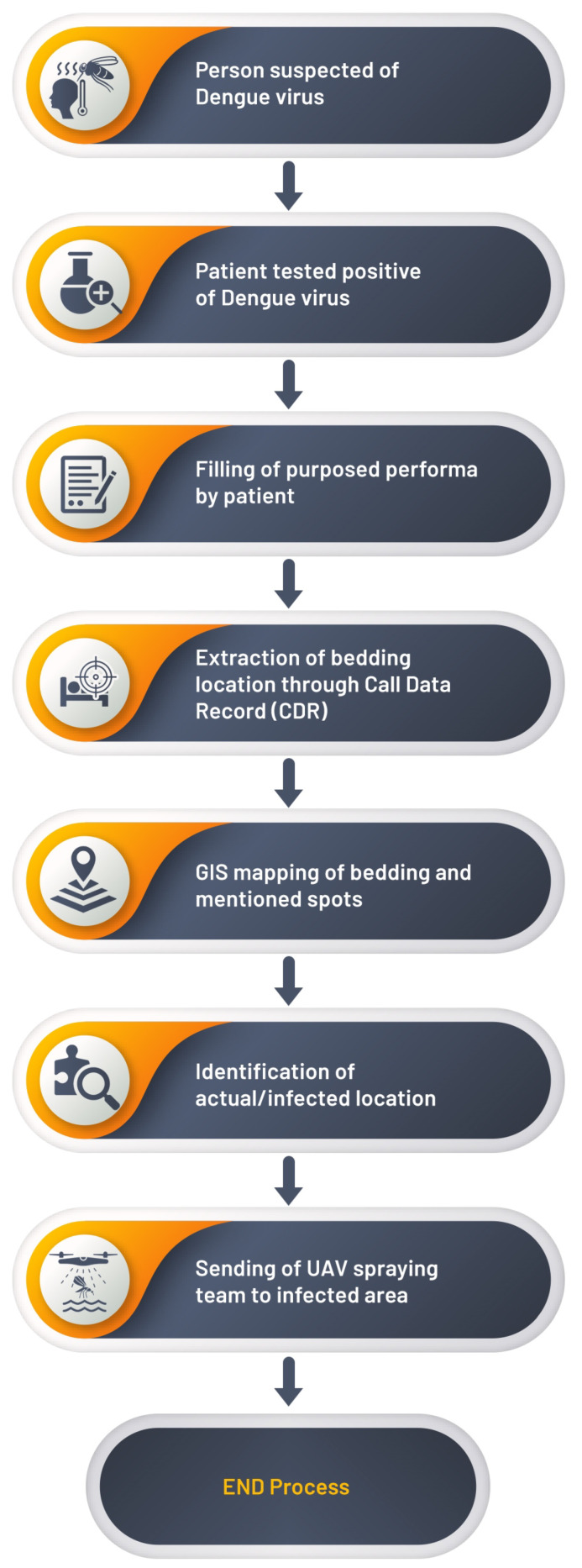
Flowchart of proposed model.

**Figure 3 micromachines-13-01702-f003:**
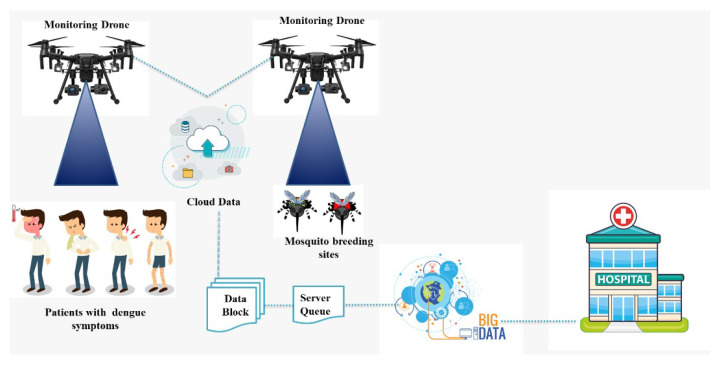
Framework graphical abstract.

**Figure 4 micromachines-13-01702-f004:**
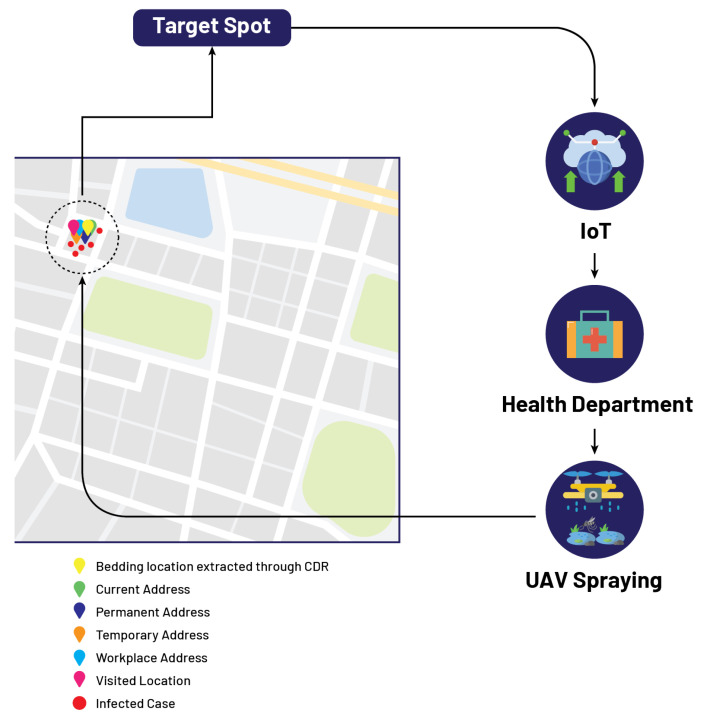
Localization of target spot based on proposed model.

**Figure 5 micromachines-13-01702-f005:**
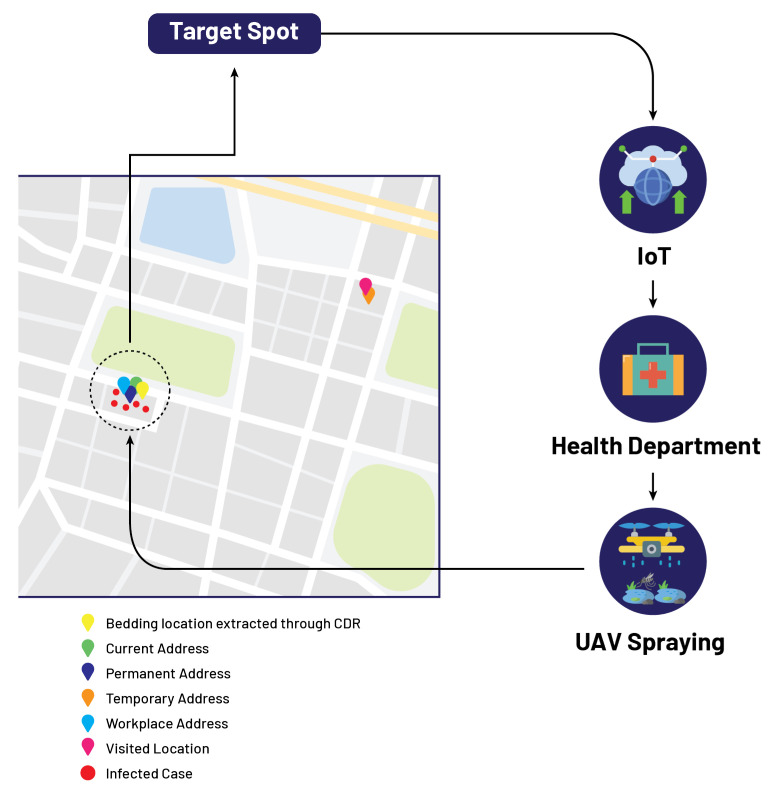
Localization of target spot based on proposed model.

**Figure 6 micromachines-13-01702-f006:**
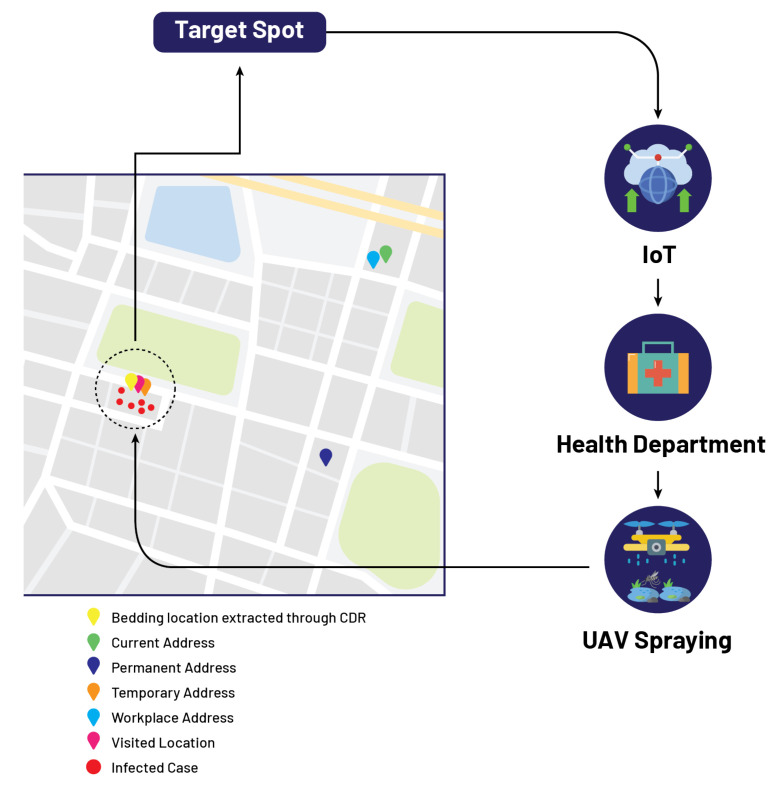
Localization of target spot based on proposed model.

**Figure 7 micromachines-13-01702-f007:**
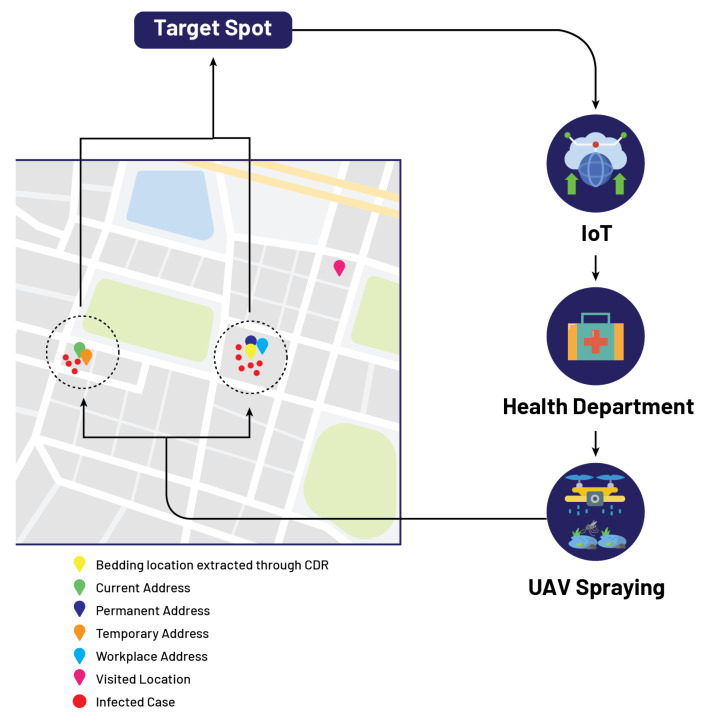
Localization of target spots based on proposed model.

**Figure 8 micromachines-13-01702-f008:**
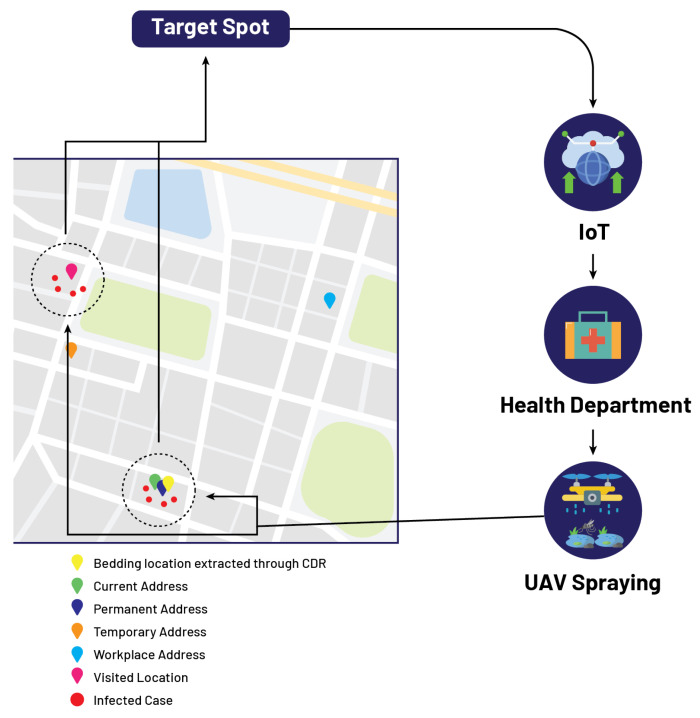
Localization of target spots based on proposed model.

**Table 1 micromachines-13-01702-t001:** Research contributions on the experimental demonstration of UAV for dengue control.

Reference	Research Focus
[68]	In this study, the authors considered several data streams to check spatial risk factors linked with dengue, associated maps gathered through UAV and the active surveillance of febrile cases. They investigated risk factors for symptomatic dengue infection in household proximity to public areas.
[69]	The prime motive of this study was to investigate the most suitable UAV to identify Ae. Aegypti habitants through aerial images of mosquito breeding sites. They used various approaches and discussed methodologies to characterize and select the most appropriate UAV for the aerial mapping of mosquito breeding sites.
[70]	The key objective of this study was to use deep learning (DL) and sensing techniques on aerial images for the detection of water tanks and swimming pools in order to control dengue.
[71]	This study proposed a novel approach for the identification of mosquito breeding sites through drone images. The proposed mechanism generates a map by capturing images of the water retention using a drone. It provides satisfactory accuracy to identify possible water retention areas and produces the final results on the basis of the effect of shadow and the drone camera tilt angle.
[72]	This study presented a UAV-based mosquito control approach that is capable of identifying mosquito breeding grounds, such as small-scale standing water bodies through drones and considering appropriate measures to stop the spread of mosquito population.
[73]	This study proposes a novel smart system for the identification of mosquito breeding habitants in man-made scenarios. To support the main objective of this study, a UAV was used to gather different configurations of aerial images to develop a database. The database was precisely annotated, and the collected images were used to test and train the proposed system. The authors used random forest as the classification algorithm for identification purposes. The overall obtained results reached a global hit rate above 99 percent for tire and water. Considering the limitation of UAV in the real-time high-resolution video scenario, the proposed system was used off-line.
[74]	This study focuses on computational mechanisms for the automatic identification of objects or scenarios considered mosquito breeding sites through drone aerial images. These mechanisms were designed through convolutional neural networks.

**Table 2 micromachines-13-01702-t002:** UAV applications.

Reference	Pandemic	Application(s)	UAVs in Dengue
[85]	COVID-19	Spray Drone	✘
[86]	COVID-19	Spray Drone	✘
[87]	COVID-19	Delivery Drone	✘
[88]	COVID-19	Delivery Drone	✘
[89]	COVID-19	Surveillance Drone	✘
[90]	COVID-19	Surveillance Sensor Drone	✘
**Our Work**	Dengue Virus	Spray Drone	✔

**Table 3 micromachines-13-01702-t003:** Proposed performa to be filled by dengue patient.

Name	Cell Number	Current Address	Permanent Address Bedding Location	Temporary Address	Workplace Address	Visited Location	Bedding Location (CDR)
ABC	033312345678	11.00°, 22.00°	12.00°, 23.00°	13.00°, 24.00°	14.00°, 25.00°	15.00°, 26.00°	11.00°, 22.00°

**Table 4 micromachines-13-01702-t004:** Information gathered from the proposed performa.

Name	Cell Number	Current Address	Permanent Address Bedding Location	Temporary Address	Workplace Address	Visited Location	Bedding location (CDR)
Mr. A	03313321024	33.48°, 73.10°	33.48°, 73.10°	33.48°, 73.10°	33.48°, 73.10°	33.48°, 73.10°	33.48°, 73.10°

**Table 5 micromachines-13-01702-t005:** Information gathered from the proposed performa.

Name	Cell Number	Current Address	Permanent Address Bedding Location	Temporary Address	Workplace Address	Visited Location	Bedding Location (CDR)
Mr. B	03312345678	33.57°, 73.06°	33.57°, 73.06°	33.99°, 71.48°	33.57°, 73.06°	33.99°, 71.48°	33.57°, 73.06°

**Table 6 micromachines-13-01702-t006:** Information gathered from the proposed performa.

Name	Cell Number	Current Address	Permanent Address Bedding Location	Temporary Address	Workplace Address	Visited Location	Bedding Location (CDR)
Mr. C	03342345678	33.47°, 73.07°	33.47°, 73.07°	31.86°, 70.90°	33.57°, 73.06°	31.86°, 70.90°	31.86°, 70.90°

**Table 7 micromachines-13-01702-t007:** Information gathered from the proposed performa.

Name	Cell Number	Current Address	Permanent Address Bedding Location	Temporary Address	Workplace Address	Visited Location	Bedding Location (CDR)
Mr. D	033312345678	34.12°, 72.46°	33.57°, 73.06°	34.12°, 72.46°	33.57°, 73.06°	34.01°, 71.52°	33.57°, 73.06°

**Table 8 micromachines-13-01702-t008:** Information gathered from the proposed performa.

Name	Cell Number	Current Address	Permanent Address Bedding Location	Temporary Address	Workplace Address	Visited Location	Bedding Location (CDR)
Mr. E	031487654321	33.57°, 73.06°	33.57°, 73.06°	34.12°, 72.46°	33.56°, 73.01°	3.57°, 73.06°	34.01°, 71.52°

**Table 9 micromachines-13-01702-t009:** Comparison of proposed model.

Tracking Capabilities	[91]	[82]	[83]	[84]	Our Work
Tracking viral infection due to dengue	✘	✘	✘	✘	✔
Tracking based on bedding location using CDRA	✘	✘	✘	✘	✔

## Data Availability

Not applicable.

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
