# Peer review of "A Privacy-Preserved Internet-of-Medical-Things Scheme for Eradication and Control of Dengue Using UAV"

_micromachines, 2022, doi:10.3390/mi13101702_

Round 1
Reviewer 1 Report
This is a well-structured paper addressing the usage of IoMT and UAV in controlling Dengue cases. However, this paper lacks some information and needs corrections, as listed below.
1) Please add the detail for the UAV used. Include a detailed description of the UAV in the model or framework section.
2) As for the references, please provide the latest references, especially on the IoMT and UAV, in controlling such cases.
3) Please include the CDRA (IoMT, UAV, GIS & Cloud) framework graphical abstract (figure representation).
4) Some grammar mistakes, please proofread the manuscript.
5) In section 5, please add the UAV implementation effect for all cases.
6) For table 7, please add the comparison for UAV implementation.
7) Please include the IoMT and UAV brief explanation in the conclusion, not just the CDRA.
Author Response
Dear sir,
Please find our response letter in the attached document.
Thank you

Reviewer 2 Report
The article introduces the call data record analysis (CDRA) for eliminating the dengue via unmanned aerial vehicles (UAVs). Although topic is quite interesting, the manuscript has to be revised based on the following comments.
Line 17: Introduction has to mention about scientific contribution of the proposed approach.
Line 132: Although authors mentioned about Algorithm 1, other equations and scientific data should be addressed in either Section 3 or Section 4.
Line 151: Information about field experiment is missing. Please design the experiment scientifically, and results can be analyzed via Statistics.
Author Response
Dear Sir,
Kindly find our response letter in he attached file.
Thank you!

Round 2
Reviewer 2 Report
The manuscript has been appropriately updated based on the previous comments.